# Determinants of the Entrepreneurial Initiative during a Pandemic: The Case of Plovdiv

Mina Nikolaeva Angelova [1,*], Daniela Dobreva Pastarmadzhieva [1] and Aleksandar Tsvetanov Naydenov [2]

1   Faculty of Economic and Social Sciences, "Paisii Hilendarski" Plovdiv University, 4000 Plovdiv, Bulgaria
2   Faculty of Applied Informatics and Statistics, University of National and World Economy, 1700 Sofia, Bulgaria
*   Correspondence: mina.angelova@uni-plovdiv.bg

**Abstract:** The COVID-19 pandemic is a trying time for both businesses and citizens. The measures and restrictions were devastating for the economy. As different countries had their strengths and challenges in dealing with the pandemic, there no unified approach applicable to every context. However, the entrepreneurial initiative is what boosts the economic development in each free market economy. The current paper's goal is to evaluate how the pandemic affects entrepreneurial initiatives and to determine the degree to which three sets of elements influence these initiatives. The scope of the research is enterprises, working in the city of Plovdiv, Bulgaria, and the focus is the entrepreneurial initiative among them. The research with the enterprises is based only on a quantitative method—a survey across a representative sample of the general population of the enterprises whose headquarters are registered in the territory of the city of Plovdiv. The representative sample was selected as a random sample of 1000 companies (with an assumed response rate of about 10%), stratified by the size of the enterprise (number of employees) and by the field of economic activity. Statistical analysis was performed using the software product IBM SPSS version 26. The results show that the personal characteristics of the respondents are more relevant to the results rather than the specifics of the enterprise. The relevance of both work experience and ownership of the enterprise as preconditions that create opportunities for entrepreneurial initiatives during the global crisis offers a further empirical contribution. A key theoretical contribution of this study lies in finding evidence that innovativeness has a significant direct effect on behavioral intention to acquire new opportunities during crisis conditions.

**Keywords:** COVID-19 and economics; entrepreneurship; innovation; Plovdiv; Bulgaria

## 1. Introduction

The COVID-19 pandemic currently affects how people behave on all continents and has implications on their economic, physical, political, social, psychological, and cultural well-being. It puts several problems on the table, but the most crucial is the approach that governments must adopt to combat the pandemic's impacts. However, the value of entrepreneurs goes beyond the impact they have on their businesses. They influence their larger communities and, in certain instances, the entire world.

Thus, the objective of the current paper is to identify the impact of the pandemic on entrepreneurial initiatives and to identify to what extent three groups of factors determine such initiatives. These groups of factors include the demographic characteristics of the entrepreneur, the attitudes toward the state's commitment to entrepreneurs' problems, and the assessment of measures introduced by the state to fight the pandemic.

The theoretical basis for entrepreneurship, as part of the social sciences, has contributions of mainstream economists and business representatives providing different ideas about the theory and practice of entrepreneurship. During this pivotal period during the pandemic, it should be emphasized that quite a few of the insights that the social sciences have already generated could be directly applied toward the entrepreneur initiative by

searching an innovative business behavior in crisis conditions than is commonly done. Businesses can strategically restructure during a crisis to leverage their advantages and get around many environmental obstacles. SMEs must adopt an intensively positive and focused approach in the face of the economic slowdown and the corresponding environmental restrictions. The conditions for supporting entrepreneurship during a crisis are especially challenging for entrepreneurs and small businesses due to the high levels of economic uncertainty created [1].

The entrepreneurial initiative and motivation can boost the Bulgarian economy, but an individual approach is needed, regarding the specifics of the context. As a result of the present empirical research, the identified characteristics of the attitudes are rather significant for the preparation of concrete measures for Bulgaria. Based on an analysis of the obtained results and proposals from businesses, the authors plan to create a strategy for dealing with crises and promoting entrepreneurship, which will be proposed to national and local governing bodies. It is of interest which of the measures provided to support businesses and employees by the state are used by the organizations. The state must establish measures to ensure not just the creation of new businesses but also their viability [2]. In the conditions of a pandemic, the provision of relief and assistance to businesses is extremely important, but it is also important to what extent the state's efforts are appreciated and approved by the organizations. The degree of success of the measures is an indicator of the strength of the state policy in the context of promoting entrepreneurship and entrepreneurial initiative. Most governments are experiencing enormous unemployment because of COVID-19. This is especially difficult in developing nations (such as Bulgaria) with weak social safety nets and high rates of unemployment and poverty before the pandemic. This circumstance can speed up the "brain drain". To solve these problems, entrepreneurship must be encouraged, as the bulk of jobs in developing nations are created by small businesses. Overall, policymakers in developing countries would be more interested in supporting the increased adoption of entrepreneurial initiatives if there is empirical evidence that the enterprises specifically benefit from the well-developed policies and strategies to encourage the establishment and development of organizations.

Located in the Balkan Peninsula, Bulgaria is a country covering an area of 110,994 km$^2$ and has a population of 6.847 million as of December 2021 [3]. Small- and medium-sized enterprises, accounting for almost 98% of all enterprises, present the backbone of the economy. Small firms can combat the negative effects of the economic downturn by utilizing techniques such as digitization, innovation, and branching out into new market niches [4]. The process of transformation of the Bulgarian economy from a planned to a market economy has been delayed due to the unstable political situation and the impossibility of undertaking large-scale reforms. The socio-historical cataclysms brought on by the country's unique geostrategic position produce a dynamic atmosphere that fosters the formation and development of the Bulgarian Entrepreneurial Initiative. In Bulgaria, the ideas of entrepreneurial spirit and entrepreneur ecosystem are frequently addressed. The creation of successful instances and effective practices, however, can only be seen in recent years. Bulgaria's economy is underdeveloped, considerably behind that of other EU nations, and lags behind other developing nations' economies [5]. Bulgaria also trails most EU member states in terms of the Global Competitiveness Index [6]. Because socialist planning and industrialization occurred simultaneously, Bulgaria has little to no experience with a market economy. The current COVID-19 crisis and government containment measures and restrictions on businesses and citizens saved lives, but they also harmed Bulgaria's economy. Several authors state that lockdowns due to pandemics have caused the worst economic recession since the great depression [7]. Additionally, it has been noted that SMEs that thrive in a downturn has a solid and long-term capital structure, stronger access to clients, and a flexible approach to strategy [8]. The declared state of emergency and the strict measures that followed greatly changed the economic life of the country. A review of the literature from previous global epidemics, as well as modeling of the potential macroeconomic consequences, shows that the effect of such a shock is different from that

observed in a financial and economic crisis. The pandemic is causing a very deep short-term shock to the economy that affects everyone, albeit through different channels and to different depths. The pandemic has heightened the need for Bulgarian enterprises to look for innovative business models and niches to fight the consequences of the deteriorating economic conditions.

To carry out an in-depth analysis of the motivation and initiative for entrepreneurship, we conducted an empirical study trying to confirm our basic hypothesis: during a pandemic, entrepreneurship is possible only if managers and entrepreneurs look for and apply innovations and innovative business models. The city of Plovdiv is the second biggest city in Bulgaria, and it has developed rapidly over the last 20 years because of the Trakia Economic Zone (TEZ). It is an industrial and commercial area and one of the biggest economic projects in Bulgaria It comprises six significant industrial zones in the Plovdiv area, totaling 1070 ha, of which 325 ha are occupied. The TEZ is home to over 180 international and Bulgarian businesses that employ over 75,000 people in the industrial sector. Over 1.1 billion euros have been invested in fixed capital in TEZ since 1995 [9]. The city of Plovdiv has placed among the top three in the category "FDI strategy" in the ranking "European cities of the future 2018/2019 (Top 10 Small European cities of the Future 2018/2019)" of the renowned British edition "Financial Times" [10]. This is due to the accomplishments of Trakia Economic Zone. In the rating of the world's media, Plovdiv and the surrounding area are ranked fourth.

A large piece of evidence from the literature shows that in the situation of crisis adopting efforts by the state in the development of strong entrepreneurial culture and competencies in entrepreneurs in developing economies contributes to increased entrepreneurial initiatives [11,12]. Furthermore, there are studies concerning the establishment of a strong entrepreneurial culture in transition economies, although with divergent findings [13,14]. This implies that the results from these studies are not only mixed, but they are also inconclusive. Most importantly, many of these studies only investigate the factors influencing the founder's departure. This finding does not give specific information on whether the entrepreneurial initiative is influenced by the state's policies. This suggests a gap in the literature about the potential heterogeneity impact of the applied measures to revive the economy. Of great importance in many areas of empirical economic research are the ability to understand or provide answers to the effect of any intervention on the entrepreneurial process. This study seeks to fill these gaps by addressing the following research questions: Are the economic measures to deal with the crisis positively evaluated by the business and are they adequate in the direction of the conditions and specifics of the enterprises? How do the entrepreneurs evaluate the efforts in applying different measures to revive the economy? Do they use the suggested programs, and if so, to what extent?

To the best of our knowledge, the entrepreneurial initiative and motivation during a pandemic is still not a well-researched area, particularly in the specifics of Bulgaria and the city of Plovdiv. Contributing to the literature by filling this gap is one of the primary motivations for this study. The empirical evidence and summarized attitudes from business representatives toward the opportunities or entrepreneurial initiative will support the policymakers to promote entrepreneurship at the national level. The authors intend to provide a practical and straightforward presentation of entrepreneurial motivation and to illustrate why the need for applying adequate state policy in Bulgaria is seemed necessary.

Regarding our main purpose, the current paper is structured as follows: following the introduction, the second part presents the literature review based upon contemporary research in the field of entrepreneurship. The third part is focused on the methodology of the research. The latter is the framework of the study and is the basis for the fourth part, which presents the empirical results, followed by a discussion of the analysis and results. The paper closes with conclusions and recommendations for future research in the field of entrepreneurship, i.e., good practices and innovative solutions applied during a pandemic.

## 2. Literature Review

### 2.1. Theoretical Framework of Entrepreneurship

A crisis is a testing time that brings to the fore the entrepreneurial abilities of the management team. On the one hand, difficult conditions lead to the bankruptcy of many enterprises, but at the same time, they provide opportunities to find niches, to look for more and more innovations, to use hidden opportunities, and to turn threats into strengths. For decades, the act of starting a firm or business while accepting all the risks in the pursuit of a profit has been the definition of entrepreneurship. It is the process of figuring out new methods to combine resources to increase the overall size of the economic pie [15]. The crucial role of entrepreneurs has not received enough attention in research on pandemic responses. Joseph Schumpeter [16] also emphasized the role of the entrepreneur as an inventor who introduces new products or new processes of production in order to effect change in an economy. The entrepreneur is a disruptive force in an economy, according to the Schumpeterian theory. In this sense, the entrepreneurship relates to establishing new and/or developing existed organization based on new business models. When a firm is referred to as "entrepreneurial" in the entrepreneurship research community, this frequently relates to entrepreneurial orientation (EO), or "what it means for a firm to be entrepreneurial at the most fundamental level" [17], p. 861. EO is regarded as a strategic construct and is connected to the strategic posture of a company (or business unit within a company) [18]. According to the definition of EO, it consists "only of continuous behavioral patterns (showing risk-taking, innovation, proactiveness, autonomy, and/or competitive aggression) whose presence permits entrepreneurship to be recognized as a defining quality of the organization" [17], p. 858. Ultimately, then, what is critical to understanding entrepreneurial initiative is a focal concern with opportunities, and the motivation that develops around that interest. Therefore, by adopting a definition of entrepreneurship as a process centrally concerned with opportunities [19], combined with the entity and general property information, a conceptual definition of entrepreneurial initiative can be produced. The history of entrepreneurship initiative demonstrates the significance and interest in the linkages between entrepreneurship initiative and outcomes such as performance and growth. Entrepreneurship initiative has a defined strategic focus.

Entrepreneurship was emphasized by Israel Kirzner [20] as a process of discovery. A person who finds previously undetected business opportunities and acts as an equalizing force is known as Kirzner's entrepreneur. Many authors further developed the meaning of entrepreneurship and the role of managing team for business survival. The book of Mintzberg [21] describes the manager's job using findings of empirical studies conducted internationally throughout many levels of management. In this sense, entrepreneurship relates to proactiveness. The specialized literature considers entrepreneurship as one of the factors that would stimulate growth [22,23]. The idea behind this approach is that the entrepreneurial initiative and entrepreneurial orientation would generate economic growth and employment, leading us out of the economic crisis caused by the pandemic.

One of the aspects of the entrepreneurship, examined in the scientific literature, is related to the personal characteristics of the entrepreneurs. The latter exploit new opportunities and are associated with disturbing the market equilibrium. They often revolutionize industries overturning long-established technologies, business models, and dominant companies [24]. To do so, they innovate and take risks. An entrepreneur is a person who acts in hazardous circumstances, or in other words, a person who buys products with a known price, to sell them with an unknown price in the future [25]. Entrepreneurship requires action—entrepreneurial action through the creation of new products/processes and/or the entry into new markets, which may occur through a newly created organization or within an established organization [26].

The proactiveness of entrepreneurs relates to applying Innovations. Lorenz and Potter [27] stated that learning organization or discretionary learning SMEs are characterized by high levels of self-planning of tasks by employees, teamwork, knowledge exchange with employees and supervisors, on-the-job training, and employee performance incentives. At a

macro level, countries with high proportions of these SMEs have higher rates of new-to-the-market innovations among SMEs and of SME innovation collaborations with other firms and organizations. Their findings point to the potential role of policies favoring administrative change in SMEs as a means of stimulating SME innovation. However, the relation between entrepreneurial characteristics, entrepreneurship education, and the intention to be entrepreneur is fundamental in the context of entrepreneurship motivation [28]. The findings of the study of Rahman et al. [29] indicate that three dimensions of entrepreneur characteristics and entrepreneurial education are positively related to the intention to be entrepreneurs. Other authors demonstrate considerable evidence for the interaction effects among financial, human, and social capital as well as among different measures of financial capital, human capital, and the fear of failure. Hanif et al. [30] investigated these effects and the personal dispositional traits on the entrepreneurial intentions among early retirees in the ICT sector of Pakistan. Following this argumentation, we included in the empirical study citizens to find dependencies and correlations among them and business representatives. The personal traits are extensively examined in Ahmed et al. [31]. Alongside the demographics, the authors examined personal characteristics such as innovativeness, need for autonomy, levels of internal locus, propensity to take risks, and stress tolerance.

Furthermore, there are numerous studies which focused on various demographic characteristics of the entrepreneurs, such as gender [32–34], age [32–35], education [32,34], work experience [31,32,34,35], family background [31,33], position [35], etc. In our study, we focused on the examination of demographic characteristics according to the availability of the data. These characteristics are age, education, and work experience. Therefore, the following hypothesis can be formulated:

**H1:** *Entrepreneurs' personal characteristics are positively correlated with the economic prospects that result from their work.*

From the authors' standpoint, as concerns the personal characteristics of the entrepreneur the largest potential for entrepreneurial initiative and motivation during the pandemic can be found among people with sufficient work experience.

### 2.2. Entrepreneurship during a Pandemic and the Authorities' Commitment

Many researchers today concentrate their efforts on addressing global crises. The role of social scientists in creating knowledge that is directly relevant to global difficulties and crises and has to be incorporated in actions taken to address them is presented in the book edited by Bartunek [36]. It advances our understanding of social processes connected to planetary catastrophes. The book also demonstrates the continuing personal development necessary to effectively address global problems by revealing interventions through persons dealing with difficulties and crises first-hand. The scientific data pertain to the multi-dimensional social dynamics at the core of these problems and the organizational actions that could address them, and it aids in our understanding of the complexity of global crises such as pandemics and climate change. Entrepreneurs carry out comparable crucial economic and societal tasks during a pandemic. Entrepreneurs can play significant roles despite operating in a setting that limits the scope and type of their entrepreneurial activity, according to Storr et al.'s discussion [37]. The position of the entrepreneur in the Republic of Croatia is examined by Radlovic et al. [38], who also pinpointed the main issues that entrepreneurs face during the coronavirus pandemic. The coronavirus pandemic has significantly changed how entrepreneurship and business activities are conducted, as shown by the research findings. Many businesses are compelled to cut pricing for their goods and services, implement corporate information technology, and invest in employee training to adapt their supply to market demands.

COVID-19 has had an extremely negative impact on entrepreneurship, as it has on other economic issues. Following the considerable decline in demand brought on by confinement and other significant limitations on travel that countries implemented to stop the spread of the disease, many businesses have failed. Due to the decreased demand,

those enterprises that continue operating do so at a lower yield, and many of them are concerned about the future due to the emergence of fresh coronavirus outbreaks brought on by the disease's increased virulence [22,23].

According to Ben Hassen [39], the pandemic demonstrated the necessity of advancing non-hydrocarbon industries by strengthening the essential tenets of the knowledge-based economy: ICT, innovation, R&D, education, entrepreneurship, and the economic and institutional framework. According to his research in Qatar, the nation has a strong foundation for converting to a knowledge-based economy. He also makes some proposals for structural reforms to promote innovation, entrepreneurship, and education in Qatar. Economic diversification may help countries reduce their exposure to recessions, market fluctuations, and technology changes, increasing their resilience to external shocks. The economic crisis created by the COVID-19 pandemic could be an opportunity to boost diversification efforts toward a knowledge-based post-oil economy.

According to Hadjitchoneva [40], the pandemic has given a new dynamic to entrepreneurship and innovations. As we are still in the pandemic, the recovery of normality in society and economics is still an ongoing process and, thus, these topics need to be examined. It is obvious that environmental hostility and proactiveness have an impact on the expansion of small- and medium-sized businesses. Numerous studies of these factors demonstrate that corporate managers should take the initiative in developing business strategies [41]. To take advantage of opportunities for business sustainability, organizations should conduct a thorough scan of their external environments. In this sense, the successful enterprises during pandemics establish innovative practices based upon managers' entrepreneurial culture. Based on empirical data, Stokke [42] determined that policies should be applied through a hands-on methodology, where the main objective is to transfer knowledge of value to the business community.

The relationship between entrepreneurial orientation (EO), strategy, organizational configuration, and firm outcomes are very important features of the EO construct. The research of McMullen et al. [43] investigated opportunity-motivated entrepreneurial (OME) activity and necessity-motivated entrepreneurial (NME) activity on 10 factors of economic freedom and gross domestic product (GDP) per capita for 37 nations. They found that both OME and NME are negatively associated with GDP per capita and positively associated with labor freedom, but that various other factors of economic freedom are uniquely related to either OME or NME. In this sense, various factors that affect entrepreneurship can be considered.

The way the national governments react and communicate with business in such a complicated situation, such as the COVD-19 pandemic, has an impact on the entrepreneurial initiative. Such statement is based on the significant of trust in the national institutions for entrepreneurship [44–46]. Especially in the context of the pandemic, the World Economic Forum draws the attention on building trust between business and governments in the context of post-pandemic world [47]. Consequently, the second hypothesis to verify is:

**H2:** *The state's and/or EU's commitment to entrepreneurs' challenges would help to advance entrepreneurship and entrepreneurial initiative.*

### 2.3. The Role of the Ecosystem and Measures Introduced by the Government

Entrepreneurship during economic uncertainty is a possible mission, because when people are not sure about the situation, they are trying to do their best and spend more efforts to look for innovative solutions. During the pandemic, entrepreneurs opened their own businesses at more than twice the rate seen in pre-pandemic times, thanks to government support programs and improved remote technology that were not available during other economic downturns such as the Great Recession [48]. Entrepreneurs in the private and non-profit sectors can assist communities in addressing public health issues. The crucial role of entrepreneurs has not received enough attention in research on pandemic responses [37]. Entrepreneurs can act as focal points for catastrophe survivors as they establish their plans for reconstruction in the context of post-disaster response and

recovery by providing necessary goods and services, restoring and rebuilding shattered social networks, and more.

The impact of the epidemic on the entrepreneur's location, the scope of the government-mandated shutdown, and the degree of in-person connection with the business all affected the degrees of uncertainty. Early-stage business owners typically saw more opportunities in the pandemic, probably because of their greater ability to pivot than those who had been in operation for more than 42 months. Additionally, early-stage business owners frequently have a more optimistic predictions for future growth.

Figure 1 highlights the total early-stage entrepreneurial (TEA) and established business ownership (EBO) rates for all 43 participating GEM economies and is based on the Global Entrepreneurship Monitor 2020/2021 study. In most economies, the ratio of early-stage entrepreneurs is higher, and they also tend to be more optimistic. This is especially valid in the context of the COVID-19 pandemic [49]. The figure presents the nascent entrepreneur (actively planning a new business) or owner-manager of a new business (within the first 42 months of starting). The framework gives one a foundation for assessing how current and future government rules and policies will affect their firm. According to studies, opportunity-driven entrepreneurship is good for one's subjective well-being. Additionally, we discover that the level of opportunity-driven and creative entrepreneurial activity is influenced by subjective well-being [50]. These findings have consequences for decision makers who want to advance both economic and subjective well-being in the country.

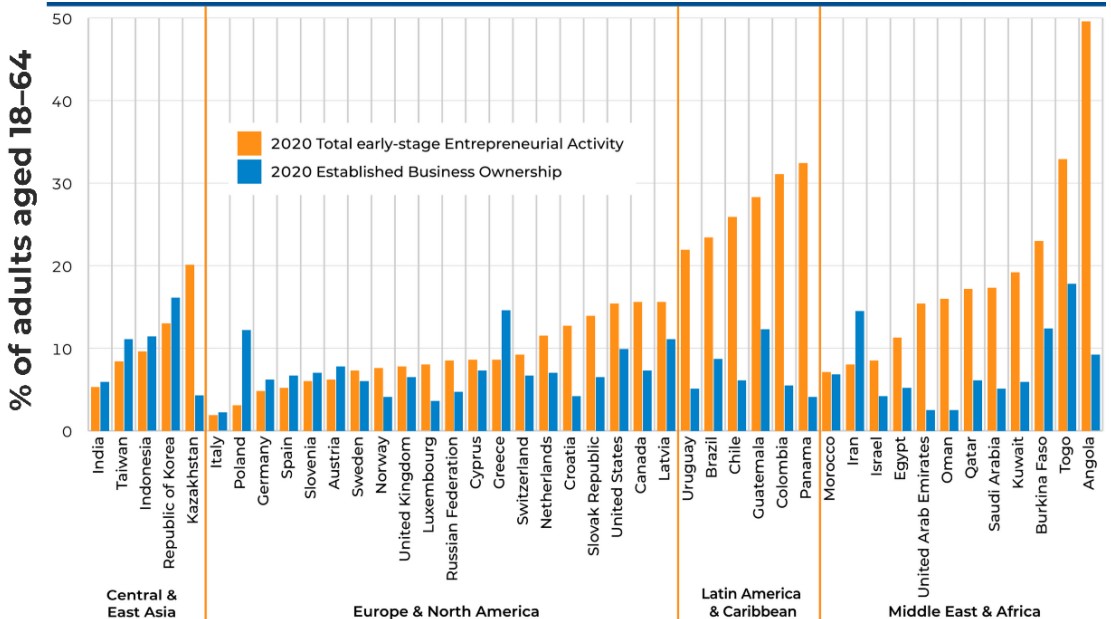

**Figure 1.** Entrepreneurial activity in 2020. Source: Global Entrepreneurship Monitor, 2020 [49].

Qudah's research [51] demonstrated the beneficial impact of entrepreneurship activities on creativity and innovation within businesses. According to him, "culture, management support, technology, strategies, and resources appeared to be the most influential aspects, in that order". For corporate transformation, change, profitability, management, and overall sustainability, innovation is a strategic tool. However, the entrepreneur's guts and desire to take the risk of making a profit or losing money due to unforeseen and unpredictable situations. Other researchers aim to demonstrate how different arrangements and characteristics of institutions can generate or mitigate uncertainty, thereby facilitating or hampering the possibilities of entrepreneurial action [52]. The entrepreneurship ecosystem refers to the coordination of institutional actors and natural persons articulated for the development of entrepreneurial projects, under the framework of public–private alliances [53]. Morales et al. [54] stated that the lack of entrepreneurial culture hampers and diminishes

the creation of new business initiatives and stress about the important role of institutions to promote entrepreneurship. The construction of the entrepreneurial ecosystem is undoubtedly the greatest challenge that the public–private institutions in every country must face. This is also valid for Bulgaria and many authors focus the attention according to the role of the so-called triangle of knowledge: science-education-business and its importance for the elaboration of entrepreneurial initiatives [55].

The COVID-19 pandemic has undoubtedly affected many economies and remains a public health issue in many areas. Therefore, it is feasible that business owners who are not adequately responsive could still have problems because of their inability to recognize opportunities. However, policymakers and scholars should, therefore, provide evidence of whether entrepreneurs were successful as a result of their responses to the pandemic and how these decisions affected the entrepreneurial activity rates of entire economies [49].

Immediately after the start of lockdowns because of the COVID-19 pandemic, the national governments introduced measures to prevent bankruptcy and support the restoration of businesses. Each of the states initiates activities in regard to its own policies and the needs of the business alongside with a plan for post-crises period [56–58]. In this sense, we can formulate the third hypothesis as follows:

**H3:** *The measures introduced by the government can boost the entrepreneurial initiative and hinder it if inadequate.*

### 3. Methodology of the Research

*3.1. Data Collection and Sampling Framework*

To carry out an in-depth analysis of the motivation and initiative for entrepreneurship in the conditions of the pandemic, two representative studies were conducted—with residents of the city of Plovdiv and with enterprises registered in the territory of the city of Plovdiv. The paper presents the results of the research with enterprises. The primary motivation of this study is to identify the impact of the pandemic on the entrepreneurial initiative and the relevance of three groups of factors, namely (1) demographic characteristics of the entrepreneur, (2) the attitudes toward the government's commitment to entrepreneurs' problems, and (3) the assessment of measures introduced by the state to fight the pandemic.

Each group of factors is related to research hypotheses defined in the second part of the research. To test our hypotheses we used the data, collected in a representative sampling survey across enterprises registered on the territory of the city of Plovdiv, Bulgaria, in the period 22 February–28 March 2022. The methodology is of a survey-descriptive type. The collected data were processed using the software product IBM SPSS version 26, and statistical analysis was made using frequencies, crosstabs, and analyses of statistical associations.

The conceptual framework from interviews to surveys is bridged by pre-testing the questionnaire with doctoral students in the field of Economics. This creates a two-phase data collection process that spans the different ideas, different methods, and different samples of respondents to create a unified picture of entrepreneurial initiative.

A content validity evaluation exercise was conducted by PhD students as the initial step in the data collection process. Hinkin and Tracey [59] outlined this approach and used it to assist choosing the right terminology and framework for their views on entrepreneurial initiative. Doctoral students were tasked with grading potential questionnaire items against dimension definitions using an analysis of variance approach. A group of questionnaire items that probed the fundamental ideas of the factors that determine entrepreneurial initiative were the initial results of this phase. The subset of items was reevaluated after looking at possible areas of item confusion and talking with faculty advisers about it. Furthermore, we tested the questionnaire with a focus group of managers and specialists of industrial enterprises in Bulgaria as expert practitioners, to test the ideas, build knowledge, and gain insightful context and nuance to the ideas. The method of expert assessment is used, and it is conducted with the support of scholars from the economic universities as consultants. A structured questionnaire was prepared and carefully administered to

gather a company's primary data. The finished questionnaire used in the second phase was created at this refining stage.

The second phase consisted of the deployment of this questionnaire to several organizations to survey entrepreneurial initiative more broadly. Surveying entrepreneurs and their employees allows for a more in-depth look at the attitudes toward entrepreneurship during the pandemic, the new business models, and innovation in the organization.

The phases for construct development and validation procedures described by MacKenzie, Podsakoff, and Podsakoff [60] had an impact on the research design. The authors address several issues they have with the conceptualization and assessment of constructs in the literature on organizational research. Their concerns include the underuse of several techniques that are useful in proving construct validity as well as the researchers' failure to sufficiently define the construct domain, specify the measurement model. A thorough process for creating a construct, from its theoretical foundation through its measurement and validation, is provided by the method described by MacKenzie et al. The conception, measure development, model specification, scale evaluation and refinement, and validation are the main steps of this process.

The research with the enterprises is based on a representative sample of the general population of the enterprises whose headquarters are registered in the territory of the city of Plovdiv. The representative sample was selected as a random sample of 1000 companies (with an assumed response rate of about 10%), stratified by the size of the enterprise (number of employees) and by the field of economic activity (level A3, according to the Bulgarian Classification of Economic Activities-2008 [61], based on NACE Rev.2), and the units in the individual strata are selected by random sampling proportional to the number of firms in the population strata. The stratification was carried out according to the data of the National Statistical Institute for the number of enterprises distributed by strata for 2020. The process of surveying the companies included in the sample was carried out in the online environment using the LimeSurvey software product, and for this purpose, an official e-mail invitation to participate in an online survey was sent to each company in the sample. The questionnaire has a volume of nine standard pages (size A4) and includes twenty-six closed questions, and the average time to complete it is 15 min. During the survey period 22 February–28 March 2022, due to the specificity of online surveys (high nonresponse rate), a total of 107 enterprises were successfully surveyed from the initial sample.

After completing the fieldwork on the survey, the data were exported from the LimeSurvey platform in two separate files in SPSS format, and then subjected to a logical review, editing, and coding of the answers to the open-ended questions.

Subsequently, the statistical processing was performed in the environment of the software product IBM SPSS version 26, to generate frequencies, crosstabs, and tests for statistical associations, including Pearson Chi-Square, Independent-Samples Mann-Whitney U Test, and Independent-Samples Kruskal-Wallis Test. Scheme 1 presents the process of implementing the selected methodology.

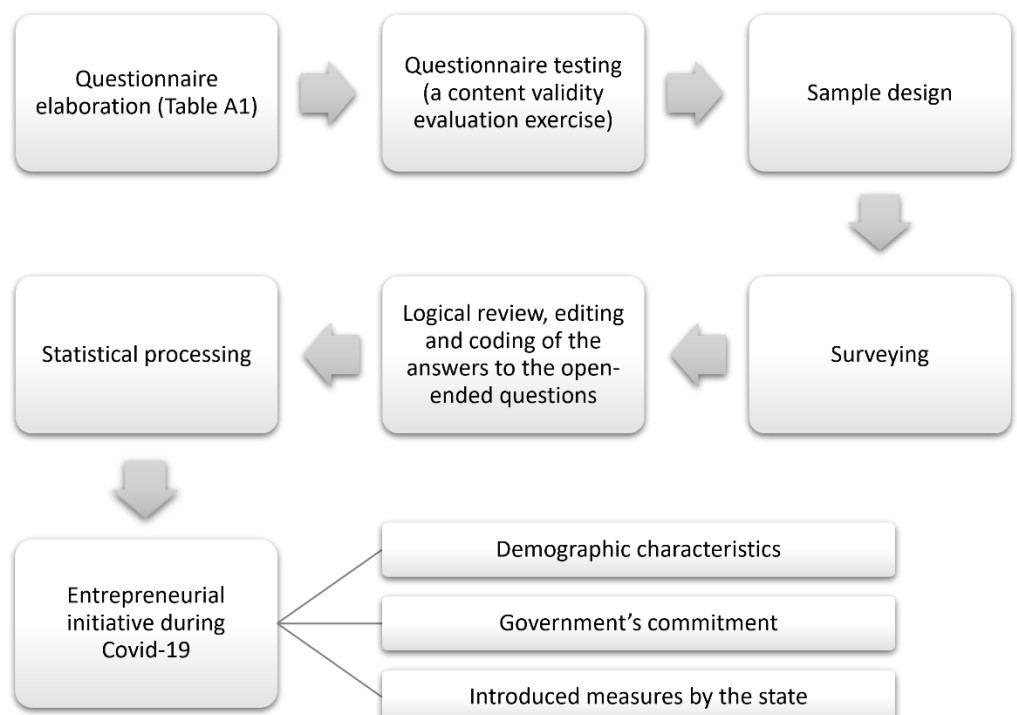

**Scheme 1.** The process of implementation of the selected methodology. Source: authors' elaboration, see Appendix A Table A1.

### 3.2. Conceptual Framework and Estimation Strategy

For the first group of factors we made cross tables with three demographic characteristics (age, education level, and work experience of the respondents) and variables, relevant to the effect of COVID-19 on entrepreneurial initiative. These variables are:

- Q1. Before the pandemic (announced in Bulgaria on 12 March 2020) did you have an intention or business plan to start your own business? This was a closed-ended question, and the respondents could give a single answer.
- Q2. In the conditions of a pandemic, is it possible to develop entrepreneurial initiative and motivation for entrepreneurship in Bulgaria? This was a closed-ended question, and the respondents could give a single answer.
- Q9. Did the pandemic period create opportunities for innovative business models in your organization? This was a closed-ended question, and the respondents could give a single answer.

The second group of factors is contained in a question Q11 and its subquestions, which are related to Bulgarian government's reaction in the pandemic. In question 11, the respondents were asked the following "On a scale of 0 to 10 (where 0—"I do not agree at all" and 10—"I strongly agree") how much you agree with the following statements concerning: political communication, unified EU support, government communication strategy, borrowing good practices from other countries, public health, etc.". There were 10 statements which they were asked to evaluate according to their opinion. We grouped the statements into three categories:

- Overall approach of the state
  - ✓ Q11.4. Messages from the politicians are contradictory and confusing, and together with a lack of unity, they create mistrust and thwart the entrepreneurial initiative.
  - ✓ Q11.5. The government does not have a communication strategy because they do not think they need one.
  - ✓ Q11.3. Political communication is effective.

- Political communication
  - ✓ Q11.6. The politicians in Bulgaria are not learning from the countries that managed the pandemic well.
  - ✓ Q11.7. The politicians in Bulgaria should borrow good practices from countries that manage the pandemic well.
  - ✓ Q11.8. It does not seem that the politicians in Bulgaria have a clear vision of how to deal with the pandemic.
- Priorities
  - ✓ Q11.9. In managing the pandemic, politicians are focusing more on public health and less on business issues.
  - ✓ Q11.10. In managing the pandemic, politicians are focusing more on business issues and less on public health.

The third group of factors focus on the assessment of measures to support the business under the pandemic. First, we aimed at identifying whether the entrepreneurs benefited from any of the state's programs (Q3. Which of the provided business and employee support measures are used in your organization?). Furthermore, we wanted to receive proposals from the respondents what else can be done to support them. Thus, we analyzed the answers of the following questions:

- Q12. In your opinion, what could the government have done better to support entrepreneurship and entrepreneurial initiative since the beginning of the pandemic until now? This question was open-ended.
- Q13. What result do you expect from the implementation of targeted support from the state and/or the EU to promote entrepreneurship and entrepreneurial initiative? This was a closed-ended question, and the respondents could give multiple answers.
- Q14. In your opinion, which of the following statements most accurately reflects your idea of support for your enterprise (or for you, if you have planned to launch your own business or startup) and improvement of the entrepreneurial environment in Bulgaria? This was a closed-ended question, and the respondents could give multiple answers.

We summarized the attitudes toward the political communication, the unified support by the institutions, the focus of politicians during the pandemic, and their impact on the entrepreneurial initiatives. It is interesting the open-ended question about suggestions from entrepreneurs in the context of the government actions and what could be done better by the politicians. The respondents assessed the truthfulness of statements that most properly captured their views on how to help their businesses and enhance the entrepreneurial climate in Bulgaria.

The following variables were used to rate the pandemic's influence on their company's growth on a five-point scale: the implementation of new production facilities, innovations (including product, market, production, marketing, management, information, etc.), external funding sources (including loans, national programs and funds, European funding, etc.), diversification (creating additional activities), the income, expenses, profit, cost price, number of workers, specialists, managers, and machines and equipment.

It is crucial to examine which measures have been implemented in businesses and to what degree they have helped respondents cope with the problems of the epidemic. The state has offered a variety of measures to support businesses and employees during the pandemic.

Finally, a test for statistical associations was performed.

To clarify the abilities of the variables of our study we added an Appendix (see Appendix A, Table A1) describing all survey questions. The quantitative research includes descriptive statistics, but also includes inferential statistical methods, such as hypothesis testing and analysis of statistical associations, including Pearson Chi-Square, Independent-Samples Mann-Whitney U Test and Independent-Samples Kruskal–Wallis Test.

## 4. Results

The analysis of the data reveals that 62.6% of the respondents occupy executive positions in the company, and 16.8% are experts in a variety of scientific subjects. In terms of direct managerial creation within the company, senior management creation accounts for 61%, middle management for 18.7%, and operative managers for 13.1%. This is consistent with the survey's inclusion of individuals who are involved in the establishment of innovative strategic plans and public policy, who oversee the organization's growth and success, and who are crucial to the development and competitiveness. These managers are firm founders or owner-managers leading companies from 6 employees to over 1200. All the managers are experienced leaders; individuals who have helped companies develop and grow. These individuals are considered experts on entrepreneurship and the culture of their organizations through their leadership and long tenure at their firms.

In addition, the crisis may have detrimental effects on businesses if it is not well managed [62], but it may also present an opportunity if decision makers assess it correctly [63]. It can be summarized that in a case of pandemic, the managerial team of experts that have work experience more than 16 years (81.3% of surveyed persons) and high educational level and competencies (more than 73.9% possess a master and doctoral degree) can overlap the traits. This include the idea of being properly managed to acknowledge the ability of organizations to respond to a crisis. They can respond to a crisis' occurrence in more or less optimal ways, even if they cannot totally prevent it or control its progress. In this sense, the significant negative effects may be related to the crisis, consistent with previous evidence suggesting a social cost to self-enhancement [64], but they may also be the product of the company's poor management [65].

According to the respondents' ratings on a scale of: (−5) to substantial negative impact; (0): no influence; and (+5): largely positive impact, the pandemic has had a significant negative impact on their business development. With a significant negative impact from the pandemic, the indices of income, expenses, profit, and cost price are anticipated to be over 40%. More than 56% of respondents believe that the pandemic has not had an impact on the number of specialists and managers working for firms, with this belief being supported by the fact that these workers are human capital and carriers of added value [66]. For more than 45% of respondents, the pandemic had no impact on their adoption of innovations, diversification, or expansion into new markets.

### 4.1. Personal Characteristics of the Entrepreneurs

In our study, we examine three personal characteristics of the entrepreneurs, which are presented in Table 1.

**Table 1.** Personal characteristics of the entrepreneurs, Plovdiv, 2022.

| Age | | Education | | Work Experience | |
|---|---|---|---|---|---|
| Up to 30 | 1% | High school or lower | 15% | 1–5 years | 1% |
| 31 to 40 | 22% | Bachelor's Degree | 12% | 6–10 years | 4% |
| 41 to 50 | 36% | Master's Degree | 67% | 11–15 years | 14% |
| 51 to 60 | 29% | PhD | 6% | 16–20 years | 21% |
| Over 60 | 12% | | | over 20 years | 61% |

Source: Based on data from the representative survey.

As concerns the respondents most of them are between 41 and 50 years old (36%), followed by the group of 51 to 60 years old (29%). Thus, most of the respondents are middle aged. Furthermore, most of them have university education with a master's degree (67%) and only 6% of them hold a PhD. The largest group of respondents have work experience over 20 years, namely 61% of them (Table 1).

With the question "Before the pandemic (announced in Bulgaria on 12 March 2020) did you have an intention or business plan to start your own business?", we aimed at identifying whether the respondents' entrepreneurial initiative was thwarted by the pandemic, keeping

in mind that all of them currently work or have their own business. However, 10% of them declared that they had a business plan, but the pandemic hindered its realization (Figure 2).

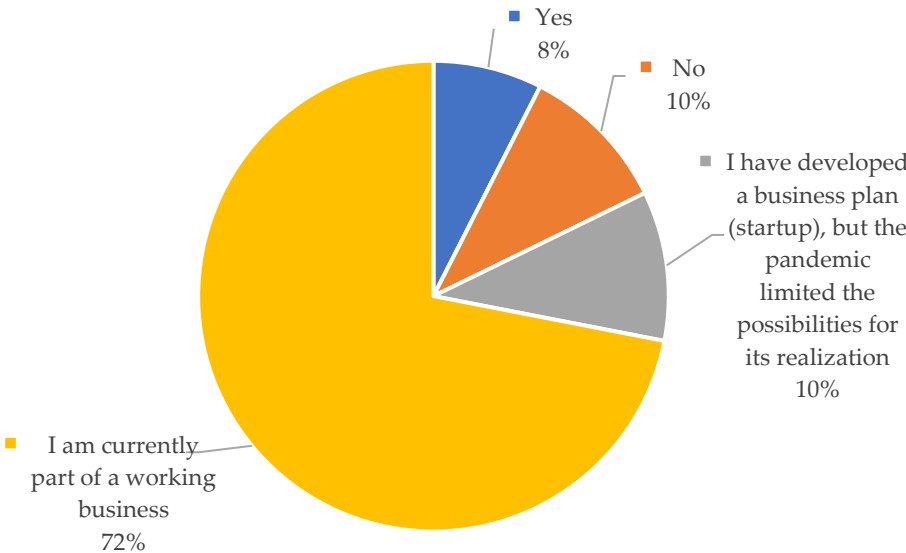

**Figure 2.** Business plans thwarted due to the pandemic, Plovdiv, 2022. Source: Based on data from the representative survey.

Figure 3 presents the results for answer "I have developed a business plan (startup), but the pandemic limited the possibilities for its realization", using demographic characteristics.

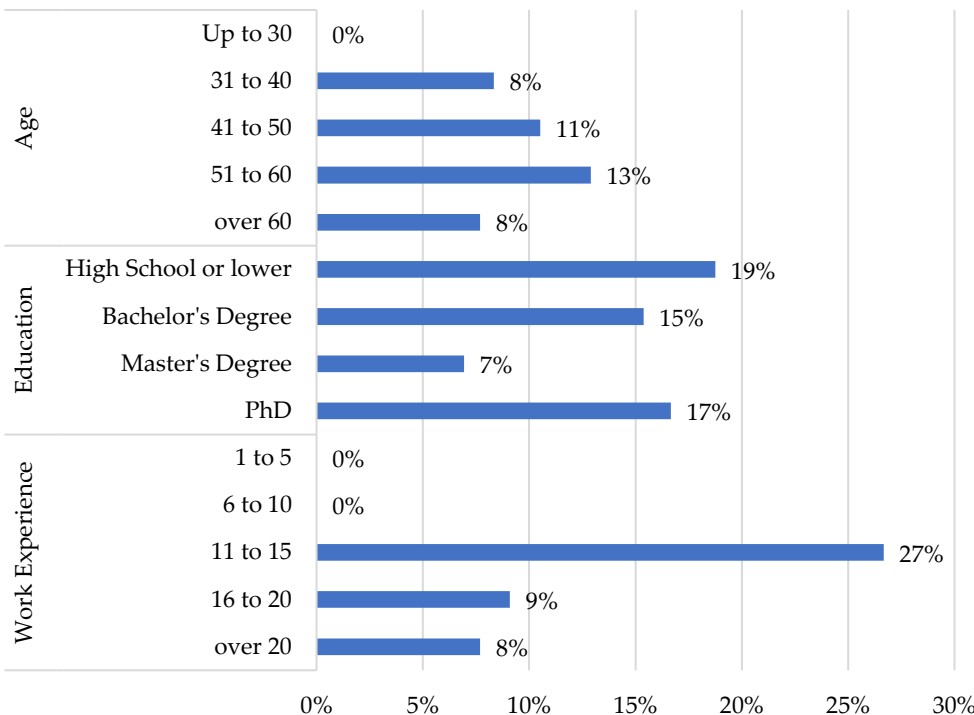

**Figure 3.** Business plans thwarted due to the pandemic according to the age, education, and work experience of the respondent, Plovdiv, 2022. Source: Based on data from the representative survey.

As concerns the demographic characteristics of the surveyed, there is a variety. The age group with the largest share of hindered business plans is of those aged 51 to 60 (13%), followed by 41 to 50 (11%). Regarding the education the most affected persons are among those with high school or lower education. Respondents with work experience 11 to

15 years are with the highest rate of thwarted business plans due to COVID-19, namely 27% (Figure 3).

In the next stage, we wanted to identify if there is a statistically significant association between the demographic characteristics and studied variables.

The results in Table 2 show that only the work experience is associated with the thwarted due to the pandemic business plans.

**Table 2.** Statistical associations between the thwarted business plans due to the pandemic and the demographic characteristics, Plovdiv, 2022.

| Variable 1 | Variable 2 | Coefficient of Significance (sig) | Statistically Significant Association (Significance Level of 5%) |
|---|---|---|---|
| Before the pandemic (announced in Bulgaria on 12 March 2020) did you have an intention or business plan to start your own business? | Age | 0.673 | No |
| | Education | 0.655 | No |
| | Work Experience | 0.006 | Yes |

Source: Based on data from the representative survey.

The respondents were asked the following question "In the conditions of a pandemic, is it possible to develop entrepreneurial initiative and motivation for entrepreneurship in Bulgaria?". It is interesting that more than half of them believe that there is such potential (Figure 4).

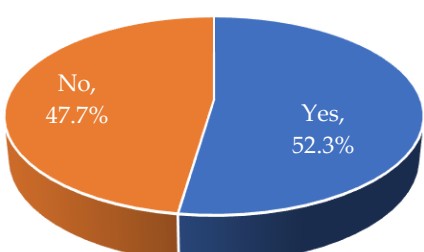

**Figure 4.** Attitudes towards entrepreneurial initiative and motivation during the pandemic, Plovdiv, 2022. Source: Based on data from the representative survey.

Next, we aimed at identifying whether there is a variety in the attitude towards the entrepreneurship depending on the selected demographic characteristics. Figure 5 presents the share of respondents who answered positively in relation to the selected demographic characteristics.

The group with the most positive attitude toward entrepreneurial initiative during the pandemic is the group of the youngest respondents, up to 30 years, as 100% of them agree with the statement. However, it seems that this attitude decreases with the age, but the oldest group also are positive minded, as 77% of them answered "Yes" to the question. As concerns the education the results show that the higher the level of education, the largest is the share of respondents, who believe that in the conditions of a pandemic it is possible to develop entrepreneurial initiative and motivation for entrepreneurship in Bulgaria. Regarding the work experience, those with the least experience are 100% positively minded about the entrepreneurial initiative during the pandemic, followed by the group of those with 16 to 20 years of experience (Figure 5).

The results indicate a certain level of association between some of the variables; thus, further statistical analyses were performed.

The results show that the variety of positive attitude towards the entrepreneurial initiative is associated only with the age of the respondents (Table 3).

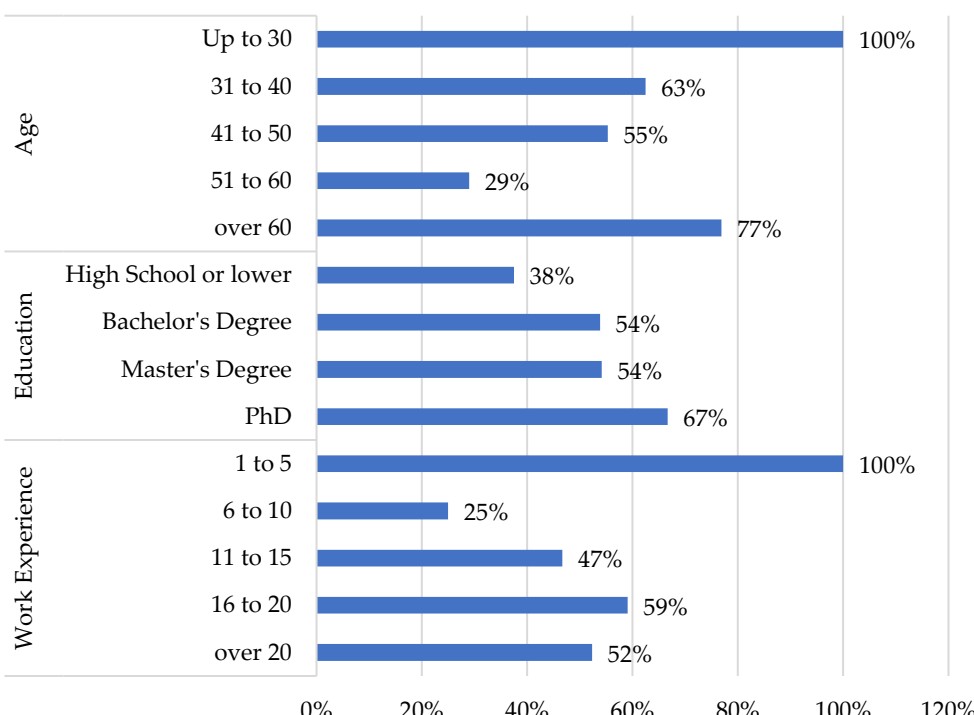

**Figure 5.** Attitudes towards entrepreneurial initiative and motivation during the pandemic according to the age, education, and work experience of the respondent, Plovdiv, 2022. Source: Based on data from the representative survey.

**Table 3.** Associations between attitudes towards entrepreneurial initiative and motivation during the pandemic and the demographic characteristics, Plovdiv, 2022.

| Variable 1 | Variable 2 | Coefficient of Significance (sig) | Statistically Significant Association (Significance Level of 5%) |
|---|---|---|---|
| In the conditions of a pandemic, is it possible to develop entrepreneurial initiative and motivation for entrepreneurship in Bulgaria? | Age | 0.10 | Yes |
| | Education | 0.434 | No |
| | Work Experience | 0.608 | No |

Source: Based on data from the representative survey.

Furthermore, we wanted not only to identify whether the respondents believe that entrepreneurship is possible during a pandemic, but if they see the pandemic period as an opportunity for innovative business models. Thus, we asked them "Did the pandemic period create opportunities for innovative business models in your organization?" (Figure 6).

As low as 16% of the respondents believe that innovative business models are possible in the conditions of pandemic. Figure 7 presents the share of respondents who replied positively across the different demographic groups.

Regarding the characteristics of the respondents, the youngest and those with the least work experience believe that the pandemic has created opportunities for innovations. Those aged 51–60 years (6%) and with work experience 6 to 10 years (0%) are least likely to think so. The majority of respondents who hold this belief have a PhD (50%), while the least educated are the least likely to hold such beliefs (0%) (Figure 7).

Again, we tested if there is a statistical association between the variables. We identified that variable 1 is positively associated with the age of the entrepreneur and their work experience (Table 4).

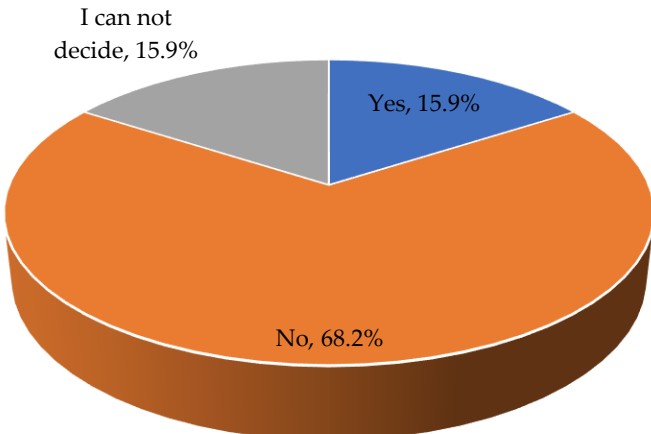

**Figure 6.** Attitudes towards the availability of opportunities for innovative business models during the pandemic, Plovdiv, 2022. Source: Based on data from the representative survey.

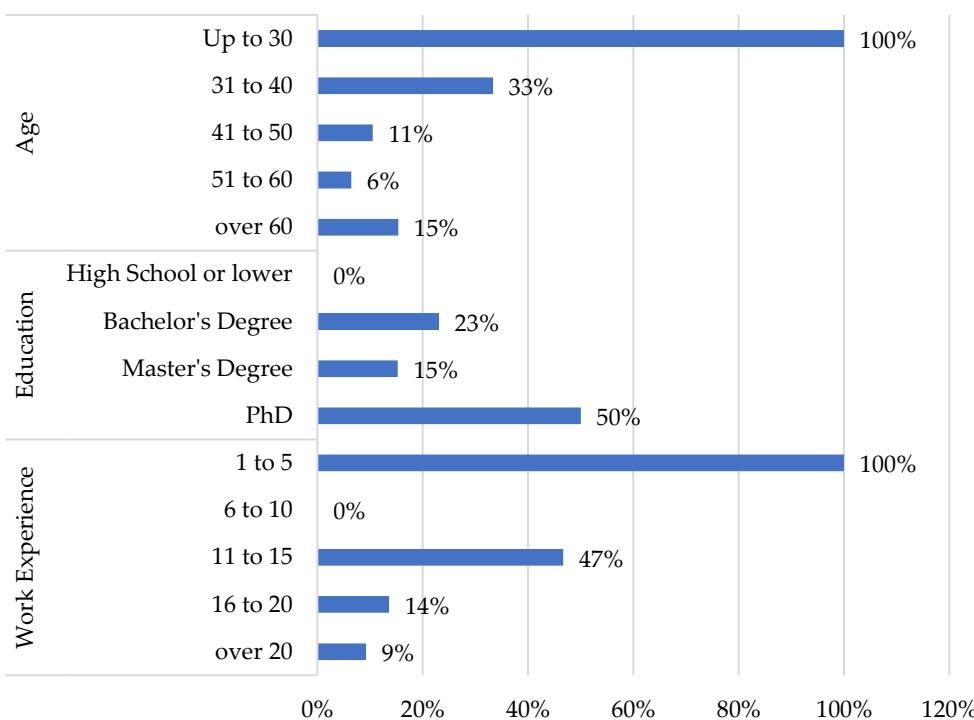

**Figure 7.** Attitudes towards the availability of opportunities for innovative business models during the pandemic according to the age, education, and work experience of the respondent, Plovdiv, 2022. Source: Based on data from the representative survey.

**Table 4.** Associations between attitudes towards the availability of opportunities for innovative business models during the pandemic and the demographic characteristics, Plovdiv, 2022.

| Variable 1 | Variable 2 | Coefficient of Significance (sig) | Statistically Significant Association (Significance Level of 5%) |
|---|---|---|---|
| In the conditions of a pandemic, is it possible to develop entrepreneurial initiative and motivation for entrepreneurship in Bulgaria? | Age | 0.019 | Yes |
| | Education | 0.364 | No |
| | Work Experience | 0.005 | Yes |

Source: Based on data from the representative survey.

### 4.2. Attitudes toward Government's Commitment to Entrepreneurs' Problems

The way the entrepreneurs perceive the behavior of the government in crisis situations determines the level of trust and entrepreneurial initiative. Thus, we aimed at identifying

whether according to the entrepreneurs the Bulgaria's government was committed enough to finding the best possible solutions in the pandemic.

The respondents were asked to determine on a scale of 0 to 10 (where 0—"I do not agree at all" and 10—"I strongly agree") how much they agree with statements concerning: political communication, unified EU support, government communication strategy, borrowing good practices from other countries, public health, etc. For the purposes of the current study, we grouped the statements in three categories and presented in Tables 5–7 the share of respondents who rather agree with the selected statement, i.e., those who selected answers 8, 9, or 10.

**Table 5.** Attitudes towards the overall approach of Bulgaria's government during the COVID-19 pandemic, Plovdiv, 2022.

| Statement | Share of Respondents Who Rather Agree |
|---|---|
| The politicians in Bulgaria do not seem to have a clear vision of how to deal with the pandemic | 63% |
| The politicians in Bulgaria should borrow good practices from countries that manage the pandemic well | 68% |
| The politicians in Bulgaria are not learning from the countries that have managed the pandemic well. | 45% |

Source: Based on data from the representative survey.

The results show that most of the entrepreneurs do not believe in the government's ability to create a proper vision for dealing with the crises and implement it. They think the politicians can learn from other countries' experience in dealing with crises and adopt a similar approach in Bulgaria. However, almost half of the respondents are convinced that the authorities do not make efforts to learn good practices (Table 5).

Furthermore, we tested some statements on the communication strategy of the Bulgaria government. The results show that according to almost 70% of the respondents, the messages of the politicians are inconsistent and lead to mistrust. Half of the entrepreneurs are convinced that the government does not even have a communication strategy. As low as 12% of the respondents assess the political communication as "effective" (Table 6).

**Table 6.** Attitudes towards the political communication of Bulgaria's government during the COVID-19 pandemic, Plovdiv, 2022.

| Statement | Share of Respondents Who Rather Agree |
|---|---|
| Messages from the politicians are contradictory and confusing, and together with a lack of unity, they create mistrust and thwart the entrepreneurial initiative | 69% |
| The government does not have a communication strategy because they do not think they need one | 50% |
| Political communication is effective | 12% |

Source: Based on data from the representative survey.

The mistrust among the entrepreneurs of the authorities also comes from the feeling that their problems are underestimated. The results, presented in Table 7, demonstrate such a perception of state policies and the balance between the public health and market economy issues. Almost half of the respondents are convinced that the focus is on the public health rather than business. Only 8% believe that business issues are priority (Table 7).

**Table 7.** Perception of state priorities in the COVID-19 pandemic, Plovdiv, 2022.

| Statement | Share of Respondents Who Rather Agree |
|---|---|
| In managing the pandemic, politicians are focusing more on public health and less on business issues | 49% |
| In managing the pandemic, politicians are focusing more onbusiness issues and less on public health | 8% |

Source: Based on data from the representative survey.

### 4.3. The Assessment of Measures Introduced by the State to Fight the Pandemic

This section discusses both the specific policies the government has implemented to encourage entrepreneurship as well as the overall atmosphere the government has built in response to the pandemic. Even if there are weakness in the political communication or overall commitment of the authorities, well targeted and implemented measures will not only support businesses but also generate more trust in the institutions.

Thus, our purpose was also to find if the enterprises benefited from the introduced measures and from which measures.

However, the results presented on Table 8 does not suggest discussing the specific measures, because more than half of the enterprises did not benefit from any of them. Furthermore, 21.5% do not cover the requirements. Different measures are used by between 1 and 10% of the respondents, which is a rather low share.

**Table 8.** Taking advantage of the COVID-19 measures for business by the enterprises, Plovdiv, 2022.

| Answer | Share |
|---|---|
| I have not benefited from the measures provided. | 52.3% |
| I do not meet the requirements for receiving support under the measures provided. | 21.5% |
| Support for small businesses with a turnover of over BGN 500,000 to overcome the economic consequences of the COVID-19 pandemic under the operational program "Innovations and Competitiveness" | 10.3% |
| 60/40 Measure | 8.4% |
| Employment for you | 5.6% |
| Support for medium-sized enterprises to overcome the economic consequences of the COVID-19 pandemic under the Operational Program "Innovation and Competitiveness" | 4.7% |
| Preferential crediting measures | 3.7% |
| Third phase of the program "Support through working capital for SMEs affected by the temporary anti-epidemic measures" | 2.8% |
| Keep Me+ | 2.8% |
| Measures for self-employed persons | 2.8% |
| Investment measures | 0.9% |
| Support for enterprises operating in one of the following codes according to the classification of economic activities (KID 2008) of the national statistical institute 55, 56, 79, 82.3, 86, 90, 93, and 96.04 (for the Tourism sector) | 0.9% |

Source: Based on data from the representative survey.

In an open-ended question, the respondents were asked to make their own proposals of what measures the government should introduce. One-third of the respondents did not reply. However, the most proposed measure is direct financial support for the enterprises. This is followed by change in the relevant legislation and noninterference in business decisions (Figure 8).

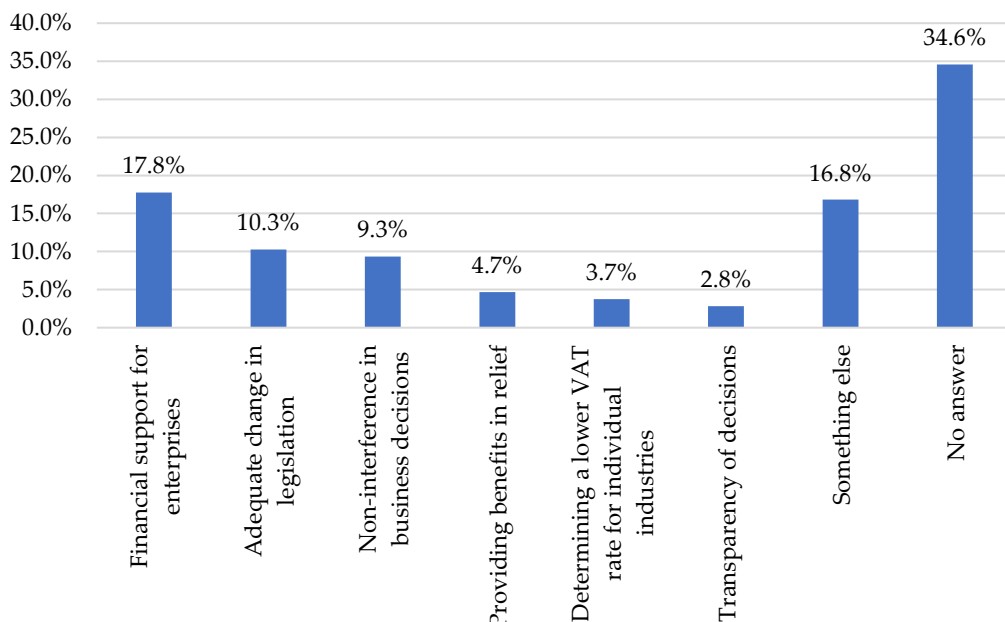

**Figure 8.** Proposals for effective measures for businesses, Plovdiv, 2022. Source: Based on data from the representative survey.

Entrepreneurs expect that a well-selected measure can lead to several positive effects, such as encouraging the entrepreneurial initiative, increasing motivation for entrepreneurship, and increasing the trust in institutions (Table 9).

**Table 9.** Expected results from the implementation of targeted support from the state and/or the EU to promote entrepreneurship and entrepreneurial initiative, Plovdiv, 2022.

| Expected Results | Share |
|---|---|
| An opportunity to significantly favor the process of creating a new business or developing an existing one | 42.1% |
| Increasing the motivation for entrepreneurship in Bulgaria | 41.1% |
| An opportunity to increase citizens' trust in institutions | 38.3% |
| Increasing business efficiency | 38.3% |
| Increase employee satisfaction and grow the business | 29.0% |
| Increasing the competitiveness of the enterprise is directly related to the measures taken by the state | 27.1% |

Source: Based on data from the representative survey.

We asked the entrepreneurs which they considered the best form of support for their enterprise and for improvement of the entrepreneurial environment in Bulgaria. We provided a list of suggestions, and they were able to select more than one answer. Almost 63% of the respondents selected "Ensuring the security of the environment" as a significant measure. It is followed by "Consistency and sustainability of policy decisions regarding business support" (48.6%) and "Financial Incentives" (43.9%) (Table 10).

**Table 10.** Support for potential measures and entrepreneurial environment improvement, Plovdiv, 2022.

| Support/Improving the Environment | Share |
| --- | --- |
| Ensuring the security of the environment | 62.6% |
| Consistency and sustainability of policy decisions regarding business support | 48.6% |
| Financial incentives | 43.9% |
| Moral and ethical rules and norms that are regulated in the legal framework | 32.7% |
| Organizational-management and educational technologies, methods, instructions, and procedures for promoting entrepreneurship (including targeted training) | 25.2% |
| Nonfinancial incentives (additional paid holidays, health insurance, provision of soft loans, employee training, food subsidies, payment of mobile operator and internet bills, flexible working hours) | 22.4% |
| Collective belief and consciousness of a community with good examples of successful entrepreneurs | 17.8% |
| Shared values, convictions, beliefs, and patterns of business behavior by all entrepreneurs | 15.0% |

Source: Based on data from the representative survey.

## 5. Discussion

The current study discusses the opportunities for entrepreneurship in the context of a crisis such as a pandemic. Nevertheless, it seems that various international situations, such as conflicts, drought, disease, etc., are affecting the business around the world. Thus, the factors contributing to entrepreneurship initiative need to be identified in order governments to introduce the best possible measures.

Based on results of the representative survey, performed across enterprises in the city of Plovdiv, we tried to identify different aspects of the motivation for entrepreneurship. We grouped the available variables into three groups of determinants to test their effect on the entrepreneurs, namely:

(1) Personal characteristics of the entrepreneurs;
(2) Entrepreneurs' perception of government's commitment to the problems of businesses during a pandemic;
(3) Assessment of the measures introduced by the state to support businesses during a pandemic.

It is inevitable that the COVID-19 pandemic hampered the economic situation around the world and the entrepreneurial initiative [1,7,37] in particular. However, it has to be noted that entrepreneurship itself may contribute to the revival of the economy [23,39,40], but first of all, the state must initiate measures to support them [22,58]. The results of our study show the extent to which the entrepreneurship was affected by the pandemic. Almost half (47.7%) of the entrepreneurs who participated in our survey disagree that in the conditions of a pandemic it is possible to develop entrepreneurial initiative and motivation for entrepreneurship. This share of the respondents identifies the pandemic as a time when the entrepreneurship is at least troubled. Almost 70% do not recognize the pandemic as a period to create opportunities for innovative business models in their organization. Such a position may mean lack of creativeness and innovativeness among this share of respondents.

The effect of demographic factors and personal traits is much examined in the scientific literature. However, here, we focused on demographics, namely age, education, and work experience. Soomro et al. [32] used almost the same demographics as in our research. The authors reached the conclusion that these characteristics are relevant for the success of the entrepreneurs. Liang et al. [35] also found age relevant. They studied the association between the median age of a country and rates of entrepreneurship, and their results show that older societies are negatively associated with entrepreneurship. As regards the age

and work experience, Chaniago reached the conclusion that "in difficult times such as the COVID-19 pandemic, having young leaders who are experienced will make it easier to achieve business success" [34], p. 399. Our research led us to similar conclusions. Among the studied demographic factors, age and work experience proved to be relevant for the entrepreneurship. Younger people are most likely to see opportunities during a pandemic. The youngest entrepreneurs also are most open to see possibilities during the pandemic rather than the older ones. However, in our results we see that as concerns the work experience, again the youngest are ready for innovations during the pandemic. Such results do not correspond to Chaniago's conclusion. This difference may be due to the overall cultural context, because Chaniago's research was performed in Indonesia. Thus, this may need further investigation.

The way the entrepreneurs perceive the national government's commitment to dealing with the pandemic and especially the problems of the business affect the trust. The latter is rather significant for the overall entrepreneurial initiative and motivation, especially in times of crises [44–47]. Positive association between the levels of trust in the government and entrepreneurial initiative was identified by Eesley and Lee [44]. A similar conclusion was reached by Çera et al. [46], who examined a context very close to Bulgarian, namely Albania. Moreover, the World Economic Forum suggests that the development in the post-pandemic period requires building trust between business and the state [47]. Our focus on the perception of the state's behavior during the pandemic aims at identifying if during this period a trust was build or if it was hampered additionally. The results give ground to conclude that the government's behavior led to lower levels of trust among entrepreneurs. The latter are not convinced that the state introduced the best possible approach in dealing with the pandemic. Furthermore, they do not think that enough efforts were made to achieve it. Additionally, communication with businesses is not seen as effective. The entrepreneurs tend to believe that the government did not even have a communication strategy. Finally, the entrepreneurs do not recognize their problems as a priority of the state. This leads us to the conclusion that entrepreneurs perceive the behavior of the government during the pandemic as poor. Such a perception hampers the level of trust, and thus, the motivation for entrepreneurship. In this context, the role of the state during the pandemic has had a negative impact on the entrepreneurial initiative and motivation.

The role of state In providing measures for entrepreneurs to recover and develop their businesses in the pandemic and post-pandemic period is inevitable [58,67,68], although some studies have reached the conclusion that "government support has no significant effect on entrepreneurial spirit" [69]. In our study, we discuss the result from the perspective that the state has a crucial role in reviving the economy and entrepreneurship. From this point of view, we aimed at finding how Bulgarian entrepreneurs assess the state's measures to overcome the COVID-19 consequences on business and their proposals for improvement. The results of the study give ground to conclude that the state must focus more on securing the entrepreneurial environment, consistent policies, and direct financial support. Block et al. [70] gave another perspective on benefiting from government support. They focused on the self-employed and believed that those who use government support and bootstrapping measure are more motivated to protect their business. This conclusion draws attention to new perspectives of our research. Our study showed that more than half of the entrepreneurs did not benefit from the government's COVID-19 measures for businesses. It is interesting to find out why not—whether it was because they did not meet the requirements, because they were refused, or because they did not even try to receive such support.

## 6. Conclusions

In conclusion, this study has successfully examined some elements that influence entrepreneurial initiative and motivation in pandemics. Since 2019, the world has been facing an unpredictable challenge, and thus, there are no pre-prepared solutions. The balance between health and economy has been hard to achieve and the states keep trying to

recover their economies. The entrepreneurial initiative and motivation can boost the latter, but an individual approach is needed due to each specific context. Thus, the identified characteristics of the attitudes are rather significant for the preparation of concrete measures for Bulgaria.

### 6.1. Theoretical and Practical Implications

The study findings provide valuable theoretical insights to the existing literature on entrepreneurship. As the SMEs present the backbone of the Bulgarian economy, the research results confirm the necessity to increase the entrepreneurial culture of business representatives. Entrepreneurial culture presents an intriguing and fertile ground for organizational research [71]. Entrepreneurial culture is an exciting phenomenon with broad implications on strategy, innovation, and the workplace environment [72]. The competitiveness of Bulgarian SMEs is still basically built by routine innovations and the profile of the innovative enterprises is low technological. One intervention that could encourage the development of this phenomenon in Bulgaria is investment in human capital and life-long learning programs [73] that focus on developing entrepreneurship competence in both active and starting professionals from all fields [74]. Based on this, we suggest increasing educational activities and initiatives in the context of a discipline called 'Innovation and Entrepreneurship', which is key to increasing entrepreneurial initiatives in Bulgaria. From the standpoint of entrepreneurship education, our combined experiences with the epidemic have given us a chance to consider and rethink how we teach and support entrepreneurs to deal with crises and uncertainty. The research of Alshebami [75] demonstrated that in times of adversity, such as during the COVID-19 pandemic and other environmental challenges, entrepreneurial resilience can act as a moderator between entrepreneurial intention and entrepreneurial self-efficacy. Entrepreneurial resilience has the potential to strengthen the relationship between entrepreneurial self-efficacy and entrepreneurial intention. There is a need for more crisis-related teaching scenarios that reflect the context-specific experiences of entrepreneurs and entrepreneurship at all stages of the entrepreneurial lifecycle [1]. The fundamental knowledge and good practices that are studied in the context of the course will enhance start-up initiatives and the search for innovative business models [76]. The mediating role of entrepreneurial resilience and the state commitment to entrepreneurs' challenges is also important, and these research findings can support policymakers in formulating relevant strategies to encourage SMEs that have been hit the worst by the COVID-19 pandemic to perform sustainably [77]. These activities will raise the competitiveness of the national economy because the entrepreneurship is a major source of innovation in a market economy [78]. From a practical point of view, since the entrepreneurial initiative is applicable during a pandemic, managers should focus on the factors which increase the innovativeness of employees. Evidence from our research suggests that modern business structures—established or start-up—face the challenge of being constantly transforming, needing to be innovative to attract a new generation of consumers and adapting to modern technologies. They all need entrepreneurs who can come up with creative solutions, and this is possible only by quality training at the national level. Policymakers must address the national strategies in accordance with the development of knowledge-based entrepreneurship and enhancement of the entrepreneurial culture of stakeholders.

Initially, we formulated three hypotheses:

- H1: Entrepreneurs' personal characteristics are positively correlated with the economic prospects that result from their work;
- H2: The state's and/or EU's commitment to entrepreneurs' challenges would help to advance entrepreneurship and entrepreneurial initiative;
- H3: The measures introduced by the government can boost the entrepreneurial initiative and hinder it if inadequate.

The first hypothesis was partially confirmed. Indeed, age is associated with the studied variable. Work experience (11–15 years), indeed, is associated with the believe that the pandemic creates opportunities for innovations, which coincide with our initial hypothesis.

The second hypothesis is confirmed, but the Bulgarian government did not show enough commitment to the problems of entrepreneurs.

The third hypothesis is also confirmed, and the entrepreneurs provided proposals how the state can rearrange its priorities to create a better environment for them.

*6.2. Limitations and Future Research Directions*

The results of this research have answered several academic calls for more up-to-date research on entrepreneurship initiatives during a pandemic. Such results are important because they show the policymakers where the potential for entrepreneurship and innovations is located. Thus, the state support measures can be targeted at these specific groups. In addition to entrepreneurial initiatives, the conceptual model and research design of this study aim to find dependencies between the possibility to develop entrepreneurial initiative during a pandemic and number of variables, such as work experience, education, number of employees, etc. The relevance of both work experience and ownership of the enterprise as preconditions that create opportunities for entrepreneurial initiatives during global crisis offers a further empirical contribution. A key theoretical contribution of this study lies in finding evidence that innovativeness has a significant direct effect on behavioral intention to acquire new opportunities during crisis conditions.

This research has the following limitations:

- The scope of the enterprises in the empirical study is narrowed according to the adopted regional principle of choice, i.e., Plovdiv, Bulgaria.
- This study focused on different segments of the Bulgarian economy.
- The scope of human resources under study is narrowed to a survey of employees in enterprises. The composition studied below was studied in terms of age structure, educational structure, work experience, and gender.
- The priority of the survey is the internal business factors that directly influence the readiness and motivation for entrepreneurship and innovativeness during a pandemic.

The current study is indicative for Bulgaria and similar representative surveys need to be performed on a national level. The results contribute to the further improvement and upgrade of the survey and creation of questionnaires for in-depth interviews. Although the research is at an early stage, this work has made important strides towards comprehensively identifying the foundational conceptual issues of entrepreneurial initiative during a pandemic and to correct incomplete and missing aspects of the construct. The result of these efforts is an opportunity for interesting future scholarship. We can identify several potential new research streams that arise from a more comprehensive understanding of motivation for entrepreneurship and the opportunities that crisis conditions create. These streams will provide significant scholarly contributions to innovative business models and niches that can be found based on innovations and to the entrepreneurship field at large. Based on an analysis of the obtained results and proposals from businesses, the team plans to create a strategy for dealing with crisis situations and promoting entrepreneurship, which will be proposed to national and local governing bodies.

**Author Contributions:** Conceptualization, M.N.A. and D.D.P.; methodology, M.N.A., D.D.P. and A.T.N.; validation, A.T.N.; formal analysis, M.N.A. and D.D.P.; investigation, M.N.A., D.D.P. and A.T.N.; resources, M.N.A.; data curation, A.T.N.; writing—original draft preparation, M.N.A. and D.D.P.; writing—review and editing, M.N.A. and D.D.P.; visualization, M.N.A. and D.D.P.; supervision, M.N.A., D.D.P. and A.T.N.; project administration, M.N.A.; funding acquisition, M.N.A. All authors have read and agreed to the published version of the manuscript.

**Funding:** The paper is part of a project № KP-06-DK-2/7/2021, funded by Bulgarian National Science Fund.

**Institutional Review Board Statement:** The study was conducted according to the guidelines of the Declaration of Helsinki, and approved by the Ethics Committee of a certified sociological agency "Noem" Ltd. The respondents confirm their participation using a Consent Form.

**Informed Consent Statement:** Informed consent was obtained from all subjects involved in the study.

**Conflicts of Interest:** The authors declare no conflict of interest.

## Appendix A

**Table A1.** Survey questionnaire details.

| Survey Question | Question Type | Measurement Level (Scale Type) |
|---|---|---|
| **Entrepreneurial initiative and activities** | | |
| Q1. Before the pandemic (announced in Bulgaria on 12 March 2020) did you have an intention or business plan to start your own business? | closed-ended, single answer | nominal |
| Q2. In the conditions of a pandemic, is it possible to develop entrepreneurial initiative and motivation for entrepreneurship in Bulgaria? | closed-ended, single answer | nominal |
| Q2.1. Please indicate which prerequisites would contribute to the development of an entrepreneurial initiative? | closed-ended, multiple choice | nominal |
| Q2.2. Please explain why you believe the development of an entrepreneurial initiative is hampered? | closed-ended, multiple choice | nominal |
| Q3. Which of the provided business and employee support measures are used in your organization? | closed-ended, multiple choice | nominal |
| Q9. Did the pandemic period create opportunities for innovative business models in your organization? | closed-ended, single answer | nominal |
| Q9.1 What innovative business models are used in your organization? | closed-ended, multiple choice | nominal |
| Q10. On a scale of 0 to 10 (where: 0—"Strongly disagre" and 10—"Strongly agre"), how much do you agree with the following statements that relate to the relationship between personal psychological aspects of the pandemic and entrepreneurial initiative: the motivation and efficiency of human resources, insufficient support from the state for business, mental health of employees, etc. | closed-ended, single answer per each row | ordinal |
| Q11. On a scale of 0 to 10 (where: 0—"I do not agree at al" and 10—"I strongly agre"), how much do you agree with the following statements concerning: political communication, unified EU support, government communication strategy, borrowing good practices from other countries, public health, etc. | closed-ended, single answer per each row | ordinal |
| Q12. In your opinion, what could the government have done better to support entrepreneurship and entrepreneurial initiative since the beginning of the pandemic until now? | open-ended | nominal |
| Q13. What result do you expect from the implementation of targeted support from the state and/or the EU to promote entrepreneurship and entrepreneurial initiative? | closed-ended, multiple choice | nominal |
| Q14. In your opinion, which of the following statements most accurately reflects your idea of support for your enterprise (or for you, if you have planned to launch your own business or startup) and improvement of the entrepreneurial environment in Bulgaria? | closed-ended, multiple choice | nominal |
| Q15. In your opinion, which of the following statements most accurately reflect your idea of the essence of entrepreneurial culture? | closed-ended, multiple choice | nominal |
| Q16. To what extent has the pandemic affected the development of your business (company's growth) according to the following indicators: the implementation of new production facilities, innovations, external funding sources, diversification, the income, expenses, profit, cost price, number of workers, machines, and equipment, etc. | closed-ended, single answer per each row | ordinal |
| Q17. What are the values you value most in the context of entrepreneurial activity? | closed-ended, multiple choice | nominal |

**Table A1.** *Cont.*

| Survey Question | Question Type | Measurement Level (Scale Type) |
|---|---|---|
| **Demographics of the enterprise** | | |
| Q4. Number of employees | closed-ended, single answer | ordinal |
| Q5. Field of economic activity | closed-ended, single answer | nominal |
| Q6. Markets at which it operates | closed-ended, single answer | nominal |
| Q7. Ownership type | closed-ended, single answer | nominal |
| Q8. Residence place type where it carries out its activities | closed-ended, single answer | nominal |
| **Demographics of the respondent** | | |
| D1. Gender | closed-ended, single answer | nominal |
| D2. Age | closed-ended, single answer | ordinal |
| D3. Educational level | closed-ended, single answer | ordinal |
| D4. Work experience | closed-ended, single answer | ordinal |
| D5. Position | closed-ended, single answer | nominal |
| D6. Management level | closed-ended, single answer | ordinal |
| D7. Duration of employment | closed-ended, single answer | ordinal |

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
