# Peer review of "Determinants of the Entrepreneurial Initiative during a Pandemic: The Case of Plovdiv"

_sustainability, doi:10.3390/su142113753_

Round 1
Reviewer 1 Report
hi, while I appreciate the authors' effeort, I have to report the following comments:
- Please describe the sample of the study and key findings in the abstract.
- There is a duplication of sentences in the introduction, the abstract example (lines 29-30), and other places in the introduction section. This is not preferred; please develop your abstract uniquely without copying from different parts of the article.
- Please specify what you mean by (private initiative) in both the abstract and the introduction section; this could mean many things, such as an individual initiative or an organization's initiative.
- Please change the word object to objective or aim. The object might have a different meaning.
- It is pretty confusing why the authors discuss information such as sample, the tool of analysis, period of conducting the survey (lines 51-57) in the introduction; this should be shifted to the research methodology section.
- The article entirely requires proofreading as many places in the essay must be grammatically corrected.
- The introduction section is poorly written; it does not give the reader a proper reason for conducting the research, a clear objective of the study, and a strong background. The entire introduction section shows only three references, and they are all related to the context of the study's demographic information. There is no literature related to entrepreneurship in various parts of the world or the context of the study. There have been many studies that have been conducted about the effect of the covid19 and entrepreneurship.
- The introduction section is not written correctly, the authors are shifting from one subject to another, there is no consistency in writing, no proper switching, it is all mess, and readers can not get what the authors are trying to say.
- The article has minimal references.
- The authors cite references irrelevant to his research even though they might discuss entrepreneurship. For example, see 204-207.
- The research methodology section contains many irrelevant and unnecessary details, such as the company name which conducted the survey and many other details. It needs to be shortened and to be written transparently. It isn't apparent.
- The study is not based on any solid theoretical background that supports the authors' arguments about the effect of covid19 on entrepreneurship.
- The literature review section is lengthy; the authors could have divided it into many areas and developed specific hypotheses accordingly.
- The authors have mixed both discussion and results. These are two separate sections. The discussion should discuss the study's findings with other studies conducted earlier. Also, the discussion section does not compare the results with a previous study. The article also does not have an implication section for the study.
- I am not sure whether the selected variables of the study can have the ability to detect the impact of the covid19 on entrepreneurship. The analysis is not precise. The article does not show the IV or DV clearly.
- The authors need to look at other articles and understand the proper way of developing articles. The authors must first explain the problem of the statement supported by the evidence or reports.
Reviewer 2 Report
Dear authors,
thanks for the opportunity to read your research, which I find very appealing. The topic of the article is up to date. Nevertheless, I'm not quite convinced with the paper. Just to point out of few concerns:
- in the introduction, you mention hypothesis (hypothesis should be clearly written and theoretically backed up),
- the topic of the article is quite board (there are a lot of different topics, but in my opinion, there is a lack of sufficient scientific explanations and background),
- the references list is in my opinion, not sufficeint,
- your quantitative research is more or less related to descriptive statistics which in my opinion is not sufficient for the journal on this level.
To sum up, the topic is extremely appealing. With additional work, this could be a great article. But at this point, I believe it is not at the level to be published in the journal.
Best regards
Reviewer 3 Report
Article
Determinants of the Entrepreneurial Initiative Under a Pan-
demic: the Case of Plovdiv
The article highlights the role of personal attributes and entrepreneurial aspirations under pandemic situation for Plovdiv- a town in Bulgaria. It points out some interesting insights but overall the article is lacking in academic qualities. The academic rigor and literature support for results, discussion are overall very weak and must be improved. These observations will help to guide the authors.
1- The authors have tried to cover the broad entrepreneurship topic under a single heading of literature. They should organize it around entrepreneurship in general and entrepreenurship under crises in separate sub-sections followed by the personal charactertistics of an entrepreneur as well as the role of ecosystem and institutions.
Entrepreneurhsip under crises situation gives rise to necessity entreprenurs as job opportunities diminish. For reference please view the literature https://doi.org/10.1016/j.jbusres.2021.08.053 ; https://doi.org/10.3390/su10124734 and https://doi.org/10.1515/erj-2018-0062 etc. Authors should also discuss from this aspect as well.
2-The need to highlight the entrepreneruial charactertisitcs and venture charactertistics under the Pandemic situation form the basis of present research but that part is not covered in detail. The research gap here exists but that is not discussed to form the basis of this study surprisingly. Authors should review it.
Similarly, the hypotheses developed here should not merely based on hearsay but they should be grounded in concrete literature. That part needs to be reviewed accordingly.
3- Methodology section speaks about two different surveys carried out for the sutdy. It does not provide any sample questions. Is it a standard questionniare and what is the criterion of choosing/asking a specific question? Is it through some panel specialists or some industry experts who approved it?
4- Some questions were open-ended while the others were closed-ended so a list of questions as Annex will be useful for the audience here.
5- The results and discussion part for this study is disappointing really. The discussion of the results is virtually missing. Not even a single result has been discussed and justified with a view to relate it with literature or the prevalent conditions and environment. It is more like a newspaper article here.
This part should be revised completely. For example if the results showed that the youth and the old-aged segment were hopeful about the opportunity arising out of crises, the discussion should focus on its implications and justifications with relation to entrepreneurhsip literature etc.
6- The research implications should be clearly provided for both research and practice. Some of the points are discussed under conclusions, they may be put together with more specific industry and research perspective.
Reviewer 4 Report
Dear Authors,
You did an excellent job presenting your paper. Congratulations!
However, in order to help you improve your work, I would like to suggest the following revisions:
1. Please revise the manuscript title as it does not clearly indicate that it is a quantitative and/or qualitative research.
2. Please rephrase and rewrite your abstract. Some information, such as the time period of data collection (line 24) and the fundamental statistical approach of the study, should be removed (line 26-27). Please limit your focus to the primary statistical technique.
3. Your background section is excessively long, vague, and unfocused. Please limit your scope. I hope you can enhance the section's writing.
4. I'm still unsure about the data collecting and analysis procedure. Please include any figures or flow charts that would help readers grasp the issue.
5. Some terminologies are used incorrectly. For example, rather than using 'representative' phrases, respondents should be used to represent the sample. In reality, for qualitative research, you should use the term "participants," but for quantitative studies, you should use "respondents."
6. When I read your manuscripts, it is evident that the study was done in two ways: qualitative and quantitative. However, there is no mention of qualitative in the abstract. As a result, I'd like you to revise the abstract, the introductory section, the methodology, and the analysis. The analysis should be separated into two parts: questionnaire distribution analysis and interview session analysis. Furthermore, the analysis of the interview session is not properly shown in the text.
7. Again, please explain the limitations of the study (lines 911-921) in the paragraph and offer elaboration for each limitation.
8. Please also forward this article to native English proof-readers who are knowledgeable on social science. I'm hoping that this writing may be condensed and published in this journal.
Good luck and best wishes!
Author Response
Dear Reviewer,
On behalf of the authors’ team, a short note to thank you for all your efforts, valuable comments, and the assessment of the transformations that we made! All comments concerning our manuscript entitled “Determinants of the Entrepreneurial Initiative Under a Pandemic: the Case of Plovdiv” were valuable to us and were very helpful for revising and improving our paper. The uploaded material is in “track changes” mode just to easily address the improvements and corrections in the text. We would like briefly to explain every point of your comments accordingly:
- Please revise the manuscript title as it does not clearly indicate that it is a quantitative and/or qualitative research.
In our manuscript, we used the term “interview” on line 402 (now 399) and also several times the term “interviewees”, which obviously led to a misunderstanding about the method we use. We use only a quantitative method – a representative survey. We underlined it as we rephrased a sentence on line 19 and 20, which now sounds this way:
The research with the enterprises is based only on a quantitative method – a survey across a representative sample of the general population of the enterprises whose headquarters are registered in the territory of the city of Plovdiv.
- Please rephrase and rewrite your abstract. Some information, such as the time period of data collection (line 24) and the fundamental statistical approach of the study, should be removed (line 26-27). Please limit your focus to the primary statistical technique.
The lines are removed.
- Your background section is excessively long, vague, and unfocused. Please limit your scope. I hope you can enhance the section's writing.
We restructured the section and added more sources according to the suggestions of the other two reviewers. As we received positive feedback for the work done, we are not sure how to proceed.
- I'm still unsure about the data collecting and analysis procedure. Please include any figures or flow charts that would help readers grasp the issue.
We created scheme 1 (line 457-460), which visualizes the process of data collection and analysis.
- Some terminologies are used incorrectly. For example, rather than using 'representative' phrases, respondents should be used to represent the sample. In reality, for qualitative research, you should use the term "participants," but for quantitative studies, you should use "respondents."
Only the term respondents is used now. We also substituted “interviewee” with “respondent”.
- When I read your manuscripts, it is evident that the study was done in two ways: qualitative and quantitative. However, there is no mention of qualitative in the abstract. As a result, I'd like you to revise the abstract, the introductory section, the methodology, and the analysis. The analysis should be separated into two parts: questionnaire distribution analysis and interview session analysis. Furthermore, the analysis of the interview session is not properly shown in the text.
In our manuscript, we used the term “interview” on line 402 (now 399) and also several times the term “interviewees”, which obviously led to a misunderstanding about the method we use. We use only a quantitative method – a representative survey. We underlined it as we rephrased a sentence on line 19 and 20, which now sounds this way:
The research with the enterprises is based only on a quantitative method – a survey across a representative sample of the general population of the enterprises whose headquarters are registered in the territory of the city of Plovdiv.
- Again, please explain the limitations of the study (lines 911-921) in the paragraph and offer elaboration for each limitation.
In our study, we aimed at clearly defining the limitations of the research also providing arguments. We tried not to make this part to extensive but to focus on the main study. Thus, the limitations are defined as follows:
- The scope of the enterprises in the empirical study is narrowed according to the adopted regional principle of their choice, respectively on the example of companies from Plovdiv, Bulgaria.
- This study is focused on different segments of the Bulgarian economy.
- The scope of human resources under study is narrowed to a survey of employees in enterprises. The composition studied below was studied in terms of age structure, educational structure, work experience, gender.
- The priority of the survey is the internal business factors that directly influence the readiness and motivation for entrepreneurship and innovativeness during a pandemic.
- Please also forward this article to native English proof-readers who are knowledgeable on social science. I'm hoping that this writing may be condensed and published in this journal.
We hope that we managed to make it sound even better now.
Again, we would like to express our gratitude to the detailed guidelines on how to improve our manuscript.
Best regards,
Reviewer 5 Report
Thank you for the opportunity to review this paper. I appreciate the Authors' contribution to the article's first version's improvement. In reference to the revised version (V2), I'd like to emphasize that the article addresses some very interesting and timely issues. The study's content is closely related to the topic specified in the title. The authors presented the findings of previous research in the field of entrepreneurial initiatives, indicating a research gap. The article is written in simple and comprehensible language. The authors have identified the research's limitations and how they can be used. The theoretical and practical implications of the presented results are an important part of the text. Despite its relatively local focus, the study may be of interest to international researchers.
However, I'd like to point out that the article interchangeably uses the terms "demographic characteristics" and "demographic indicators" (eg lines: 456, 591, 604). Because these are not unambiguous terms, I recommend using the phrase "demographic characteristics" in the context of the study under consideration.
Author Response
Dear Reviewer,
On behalf of the authors’ team, a short note to thank you for all your efforts and valuable comments. All comments concerning our manuscript entitled “Determinants of the Entrepreneurial Initiative Under a Pandemic: the Case of Plovdiv” were valuable to us and were very helpful for revising and improving our paper. The uploaded material is in “track changes” mode just to easily address the improvements and corrections in the text. We would like briefly to explain every point of your comments accordingly:
- The article interchangeably uses the terms "demographic characteristics" and "demographic indicators" (eg lines: 456, 591, 604). Because these are not unambiguous terms, I recommend using the phrase "demographic characteristics" in the context of the study under consideration.
We corrected the paper using only the phrase “demographic characteristics”.
Again, we would like to express our gratitude to the detailed guidelines on how to improve our manuscript.
Best regards
Round 2
Reviewer 1 Report
Dear authors
Thanks for your amendments; I am happy to see these changes. Further, the article deals with entrepreneurial activities during covid19 and other adversities. Thus I still strongly recommend as the article deals with entrepreneurship during covid19 and misfortune, so I suggest adding the following sources as these are fundamental articles; they show how individuals develop resilience during covid19 and what could be the key factors affecting their sustainability.
https://doi.org/10.1037/0022-3514.88.6.984
https://doi.org/10.3390/su14137604
Bonanno, G.A., Rennicke, C., & Dekel, S. (2005). Self-enhancement among high-exposure survivors of the September 11th terrorist attack: Resilience or social maladjustment? Journal of Personality and Social
Psychology, 88(6), 984–998.
https://doi.org/10.3390/su141710689
kind regards
Author Response
Dear Reviewer,
On behalf of the authors’ team, a short note to thank you for all your efforts and valuable comments. All comments concerning our manuscript entitled “Determinants of the Entrepreneurial Initiative Under a Pandemic: the Case of Plovdiv” were valuable to us and were very helpful for revising and improving our paper. Based on your suggestions, we tried to apply all of them in the text focused on increasing the quality of the paper. The uploaded material is in “track changes” mode just to easily address the improvements and corrections in the text. We would like briefly to explain every point of your comments accordingly:
- https://doi.org/10.1037/0022-3514.88.6.984 Bonanno, G.A., Rennicke, C., & Dekel, S. (2005). Self-enhancement among high-exposure survivors of the September 11th terrorist attack: Resilience or social maladjustment? Journal of Personality and Social Psychology, 88(6), 984–998.
We added the paper in the reference list. The mentioned research is a fundamental source connected with the consistent with previous evidence suggesting a social cost to self-enhancement in crisis conditions. Please check reference:
[64] G. A. Bonanno, C. Rennicke, and S. Dekel. Self-Enhancement Among High-Exposure Survivors of the September 11th Terrorist Attack: Resilience or Social Maladjustment? Journal of Personality and Social Psychology, 88(6), 984–998. 2005. https://doi.org/10.1037/0022-3514.88.6.984
- https://doi.org/10.3390/su14137604
- https://doi.org/10.3390/su141710689
Thank you for your suggestions. We included both articles in our research as these are fundamental papers that show how individuals develop resilience during Covid19 and what could be the key factors affecting their sustainability.
[75] A. S. Alshebami, Psychological Features and Entrepreneurial Intention among Saudi Small Entrepreneurs during Adverse Times. Sustainability 2022, 14, 7604. https://doi.org/10.3390/su14137604.
[77] A.H.A. Seraj, S.A. Fazal, and A.S. Alshebami, Entrepreneurial Competency, Financial Literacy, and Sustainable Performance—Examining the Mediating Role of Entrepreneurial Resilience among Saudi Entrepreneurs. Sustainability 2022, 14, 10689. https://doi.org/10.3390/su141710689.
Again, we would like to express our gratitude to the detailed guidelines on how to improve our manuscript.
Best regards
Reviewer 2 Report
Dear Authors,
I appreciate your efforts in re-writing the article. Overall the article is now better. Still I have some very important concerns because of which I consider the article not suitable for publication in journal at this level.
Your article is in my opinion neither quantitative, neither qualitative. Let me explain – for quantitative studies at this level, at least from my opinion, showing descriptive statistic and correlation coefficient or t-test is not sufficient. Articles usually includes structural equation modeling or at least regression analysis, and empirically tests developed model. There are also limitations from the questionnaire – in the article sources for measurement scales aren’t presented, so in such way all findings could be endangered.
Altogether, in my opinion article is based on too much general description, and to less on scientific contribution.
Wish you all the best with your study,
Best regards
Author Response
Dear Reviewer,
On behalf of the authors’ team, a short note to thank you for all your efforts and valuable comments. All comments concerning our manuscript entitled “Determinants of the Entrepreneurial Initiative Under a Pandemic: the Case of Plovdiv” were valuable to us and were very helpful for revising and improving our paper. Based on your suggestions, we tried to apply all of them in the text focused on increasing the quality of the paper. The uploaded material is in “track changes” mode just to easily address the improvements and corrections in the text.
We’d like to clarify the following aspects of our research, concerning the article:
- The questionnaire we have used (see ANNEX I) consists mainly of questions measured on the nominal scale and rarely on the ordinal one. Although representative (random), the sample size is relatively small (n=107 respondents).
- Considering the questionnaire specifics and the sample size limitations we provided mainly descriptive and inferential statistics because the usage of advanced methods, such as structural equation modeling, requires “powerful” measurements (ratio scales) and large sample sizes.
- We believe that straightforward and understandable results, which are usually obtained by descriptive and basic inferential statistics, are far more practical and useful for the real-world research findings than the ambiguous outcomes from the sophisticated modelling.
Again, we would like to express our gratitude to the detailed guidelines on how to improve our manuscript.
Best regards
Reviewer 3 Report
The authors have improved the manuscript as compared to previous version but I still believe it is not up to the mark to be considered for a journal of this rank.
The major problem is their hypotheses structure and presentation. They are trying to draw big conclusions/recommendations on the basis of limited survey opinion from general respondents. In my opinion, the qualitative nature of study would better suit their arguments drawn on the basis of specialized field experts. The hypotheses claimed here especially hyp-2 and hyp-3 lack academic rigor and are presented with little literature and evidential support. The questionnaire/survey items are also general statements without proven psychometric testing and qualities therefore I personally feel that the methodology is also flawed.
A quantitative nature of study needs to follow certain standards which are missing. I would recommend to the editor to reject this work in its current form.
Author Response
Dear Reviewer,
On behalf of the authors’ team, a short note to thank you for all your efforts and valuable comments.
Best regards
Reviewer 4 Report
This manuscript, in my opinion, is too vague. It demonstrates inadequate study designs, poor methodological details, poor writing quality, and a weak study rationale. It is suggested that some of the items be translated using a Table.
Round 3
Reviewer 2 Report
Dear Authors,
I appreciate your efforts in revising the paper and for your comments, but I find it very hard to agree with them. My personal viewpoint is strongly related to quantitative (multivariate) research methods – i.e. regression analysis, SEM, etc. In my opinion, descriptive statistics (charts, frequency tables) and some basic statistics (correlation – many variables can be correlated with each other, but this doesn’t provide any real scientific value) can provide important insights into the sample but are definitely not enough for serious scientific conclusions. In addition, your H2 and H3 are poorly designed from a statistical viewpoint. As you pointed out you have a limitation in your article, as the majority of variables are not measured as numeric variables, which limits the possible methods that you can use – due to this there is no possibility to improve the article from that viewpoint.
Again, thanks for your effort, but unfortunately quantitative part of the study and the methods used aren’t enough for strongly supported scientific conclusions.
Hope you understand my viewpoint. Wish you all the best with your research.
Best regards
Reviewer 3 Report
authors have improved the manuscript as compared to previous version but I still believe it is not up to the mark to be considered for a journal of this rank.
The major problem is their hypotheses structure and presentation. They are trying to draw big conclusions/recommendations on the basis of limited survey opinion from general respondents. In my opinion, the qualitative nature of study would better suit their arguments drawn on the basis of specialized field experts. The hypotheses claimed here especially hyp-2 and hyp-3 lack academic rigor and are presented with little literature and evidential support. The questionnaire/survey items are also general statements without proven psychometric testing and qualities therefore I personally feel that the methodology is also flawed.
A quantitative nature of study needs to follow certain standards which are missing. I would recommend to the editor to reject this work.